# SARS-CoV-2 infection and transmission in the North American deer mouse

Bryan D. Griffin[1], Mable Chan[1], Nikesh Tailor[1], Emelissa J. Mendoza[1], Anders Leung[1], Bryce M. Warner[1,2], Ana T. Duggan [3], Estella Moffat[4], Shihua He[1], Lauren Garnett[1,2], Kaylie N. Tran[1], Logan Banadyga [1], Alixandra Albietz[1], Kevin Tierney[1], Jonathan Audet[1], Alexander Bello[1], Robert Vendramelli[1], Amrit S. Boese [1], Lisa Fernando[1], L. Robbin Lindsay[1,5], Claire M. Jardine[6], Heidi Wood[1], Guillaume Poliquin[2,7,8], James E. Strong[1,2,7], Michael Drebot[1,2], David Safronetz[1,2], Carissa Embury-Hyatt[4] & Darwyn Kobasa [1,2✉]

Widespread circulation of SARS-CoV-2 in humans raises the theoretical risk of reverse zoonosis events with wildlife, reintroductions of SARS-CoV-2 into permissive non-domesticated animals. Here we report that North American deer mice (*Peromyscus maniculatus*) are susceptible to SARS-CoV-2 infection following intranasal exposure to a human isolate, resulting in viral replication in the upper and lower respiratory tract with little or no signs of disease. Further, shed infectious virus is detectable in nasal washes, oropharyngeal and rectal swabs, and viral RNA is detectable in feces and occasionally urine. We further show that deer mice are capable of transmitting SARS-CoV-2 to naïve deer mice through direct contact. The extent to which these observations may translate to wild deer mouse populations remains unclear, and the risk of reverse zoonosis and/or the potential for the establishment of *Peromyscus* rodents as a North American reservoir for SARS-CoV-2 remains unknown.

[1] Zoonotic Diseases and Special Pathogens Division, National Microbiology Laboratory, Public Health Agency of Canada, Winnipeg, MB, Canada. [2] Department of Medical Microbiology and Infectious Diseases, College of Medicine, Faculty of Health Sciences, University of Manitoba, Winnipeg, MB, Canada. [3] Science Technology Cores and Services, National Microbiology Laboratory, Public Health Agency of Canada, Winnipeg, MB, Canada. [4] National Centre for Foreign Animal Disease, Canadian Food Inspection Agency, Winnipeg, MB, Canada. [5] Department of Entomology, University of Manitoba, Winnipeg, MB, Canada. [6] Department of Pathobiology, Canadian Wildlife Health Cooperative, Department of Pathobiology, University of Guelph, Guelph, ON, Canada. [7] Pediatrics & Child Health, College of Medicine, Faculty of Health Sciences, University of Manitoba, Winnipeg, MB, Canada. [8] Office of the Scientific Director, National Microbiology Laboratories, Public Health Agency of Canada, Winnipeg, MB, Canada. ✉email: darwyn.kobasa@canada.ca

Severe acute respiratory syndrome coronavirus 2 (SARS-CoV-2) is the causative agent of coronavirus disease 2019 (COVID-19), an acute respiratory illness that is presumed to have emerged in late 2019[1] as a result of zoonotic spillover from an animal reservoir, currently believed to be the horsehoe bat (genus *Rhinolophus*) with a possible role for an intermediary host[2]. The emergence of SARS-CoV-2 led to a global pandemic associated with millions of infections, hundreds of thousands of deaths, and severe social and economic disruption.

SARS-CoV-2, a positive-sense RNA virus of the betacoronavirus genus, is closely related to the high-consequence respiratory viruses, SARS-CoV and Middle East respiratory syndrome coronavirus (MERS-CoV). While the majority of SARS-CoV-2 infections result in asymptomatic[3,4] or mild respiratory disease[5,6] some individuals progress to more severe disease that can result in admittance to an intensive care unit (ICU) and/or the need for oxygen therapy or invasive mechanical ventilation and can result in serious multi-organ sequelae and death. The severity of disease and likelihood of mortality often correlates with advanced age, compromised immune status, and the pre-existence of certain cardiovascular, pulmonary, and metabolic comorbidities[5,7]. Human-to-human transmission of SARS-CoV-2 is believed to occur primarily through short-range respiratory droplets and aerosols emitted by infected individuals during coughing, sneezing, speaking, breathing, or as a result of aerosol-generating medical procedures[8]. Individuals ranging from pre-symptomatic, asymptomatic, and mildly symptomatic to fulminant symptomatic are thought to be capable of transmitting SARS-CoV-2[9].

Like the other nonseasonal human coronaviruses, SARS-CoV[10] and MERS-CoV[11], SARS-CoV-2 is believed to be of bat origin[2,12] although the related viruses have also been identified in wild pangolins[13]. Examinations of amino acid sequences of the SARS-CoV-2 receptor, angiotensin-converting enzyme 2 (ACE2), from various vertebrate species, predict that additional wild animal species may be susceptible[14]. Several experimental animal models of SARS-CoV-2 infection have been reported including transgenic mice that express human ACE2 (hACE2)[15] and mouse-adapted SARS-CoV-2[16], as well as tree shrews[17], hamsters[18,19], ferrets[20,21], fruit bats[22], rhesus macaques[23], cynomolgus macaques[24], marmosets[24], and African green monkeys[25] that are innately susceptible to infection with wildtype SARS-CoV-2. Experimental transmission by both direct and indirect contact has been reported for hamsters[18,19], ferrets[21], and domestic cats[20,26]. Last, there have been documented instances of unintentional human-to-animal transmission (zooanthroponosis) with several species of captive or domesticated animals including tigers[27], mink[28], cats[20,29], and dogs[30]. Widespread circulation of SARS-CoV-2 in humans has raised concerns about the theoretical risk of reverse zoonosis events with wildlife[31], potentially seeding new host reservoir species and geographic regions in which SARS-CoV-like coronaviruses have not historically been endemic.

*Peromyscus* species rodents, including the deer mouse (*Peromyscus maniculatus*), as members of the *Cricetidae* family of rodents, are closely related to the SARS-CoV-2-susceptible Syrian hamsters (*Mesocricetus auratus*)[18,19]. Further, *Peromyscus* species rodents are susceptible to persistent infections with and are the natural host reservoirs of several important zoonotic pathogens, including *Borrelia burgdorferi*, the causative agent of Lyme disease[32], deer tick virus (DTV)[33], and Sin Nombre orthohantavirus (SNV)[34,35].

In this work, we demonstrate that adult deer mice are susceptible to experimental infection with a human isolate of SARS-CoV-2. SARS-CoV-2 infection resulted in asymptomatic infection or mild disease with lesions limited to mild lung pathology, despite a high viral burden and elevated levels of inflammatory cytokines in the lungs and seroconversion. Further, we show that infected deer mice are capable of transmitting SARS-CoV-2 to naive deer mice through direct contact when co-housed.

## Results

Deer mice and closely related white-footed mice (*Peromyscus leucopus*) are members of the *Cricetidae* family of rodents (Fig. 1a) that are widely distributed across North America (Fig. 1b). We examined deer mouse ACE2 for congruence with human ACE2 (hACE2) at key amino acid residues known to confer efficient binding to the SARS-CoV-2 Spike (S) (Fig. 1c) and came to a conclusion consistent with recent reports that deer mice were likely to be susceptible hosts[14]. We found that deer mouse ACE2 differed from hACE2 S-contacting residues at four locations and that the amino acid differences were unlikely to have a detrimental effect on S binding efficiency[36]. We further found that ACE2 belonging to the white-footed mouse had in common all but three hACE2 S-contacting residues, and white-footed mouse ACE2 has since been shown to bind S and confer entry to SARS-CoV-2 into Hela cells[14].

Here, eight to thirty two-week-old male or female deer mice (*P. maniculatus rufinus*; in-house colony) were inoculated with $10^5$ TCID$_{50}$ of SARS-CoV-2 by an intranasal route (i.n.) and monitored daily for clinical signs and weight loss for 21 days or necropsied at 2 and 4 days post infection (dpi). A table outlining the experimental design is provided in Supplementary Table 1. SARS-CoV-2-exposed deer mice seldom displayed clinical signs (occasionally presenting with ruffled fur), and no animals succumbed to infection (Fig. 2a) or lost weight (Fig. 2b) at any point postinfection with no differences noted related to age. At 2 dpi, high levels of viral RNA (vRNA; ~$10^8$–$10^{10}$ genome equivalents/g) were detected in the nasal turbinates and lungs, and infectious virus (up to $3.3 \times 10^7$ TCID$_{50}$/g and $1.5 \times 10^7$ TCID$_{50}$/g) was detected in the nasal turbinates and lungs (Fig. 2c, d). At 4 dpi infectious virus detected in the nasal turbinates had declined by several logs while lungs titers had declined to a lesser extent (ranging from $3 \times 10^3$ to $1.4 \times 10^5$ TCID$_{50}$/g and $6.8 \times 10^4$ to $3.4 \times 10^6$ TCID$_{50}$/g, respectively). Age did not make a significant contribution to the viral burden detected in the tissues (Supplementary Fig. 1). At 2 and 4 dpi high levels of vRNA (~$10^5$–$10^8$ genome copies/g) were detected in the small intestine and colon tissues (Fig. 2e, f), and infectious virus was present at a low level. To monitor viral shedding, oropharyngeal and rectal swabs were collected every 2 days from 2 to 8 dpi (Fig. 2g, h). High levels of vRNA (~$10^5$–$10^7$ genome copies/ml) were detected on 2 and 4 dpi in oropharyngeal swabs, and both vRNA and infectious virus peaked at 2 dpi and declined thereafter (Fig. 2g). Moderate levels of vRNA (~$10^3$–$10^5$ genome copies/ml) were detected on 2 and 4 dpi in rectal swabs, and vRNA peaked at 2 dpi and declined thereafter (Fig. 2h). Infectious virus in rectal swabs was detected on 2, 4, 6, and 8 dpi at or near the limit of detection (Fig. 2h). Viral RNA was detected in nasal washes at 6 dpi in the four deer mice for which samples were obtained (~$10^5$–$10^6$ genome copies/ml) (Fig. 2i). An additional experimental group of deer mice ($n = 5$) were inoculated i.n. with $10^6$ TCID$_{50}$ of SARS-CoV-2. A table outlining the experimental design is provided in Supplementary Table 1. In this group vRNA was detected in urine samples at 6 dpi in one deer mouse (~$10^4$ genome copies/ml) (Fig. 2j) and feces at 4 dpi in all deer mice (~$10^4$–$10^6$ genome copies/g) (Fig. 2k). Viral RNA in blood was detected on 1 dpi, indicating transient viremia that declined in most animals by 3 dpi (Fig. 2l). Studies to examine the vectorial capacity of selected tick species such as *Ixodes scapularis* for SARS-CoV2 may be warranted. An additional experimental group of deer mice ($n = 4$) was inoculated i.n. with $10^5$ TCID$_{50}$ of SARS-CoV-2, and

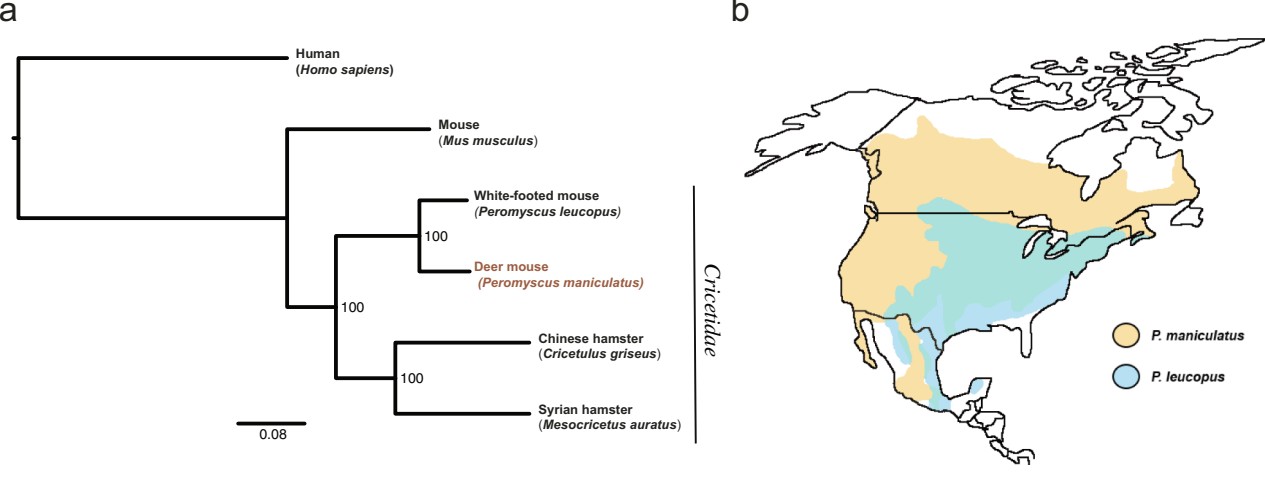

**Fig. 1 Deer mouse range and predicted susceptibility to SARS-CoV-2. a** Phylogeny showing the relationships among selected members of the *Cricetidae* family with mice and humans. **b** Geographical distribution of deer mice (based on data from Hall, 1981)[43]. **c** Alignment of human ACE2 (hACE2) amino acid residues known to confer efficient binding of the RBD of SARS-CoV-2 spike with the corresponding ACE2 amino acid residues from selected members of the *Cricetidae* family and other naturally or experimentally susceptible host species. The physiochemical properties of the amino acids residues are indicated: nonpolar (yellow), polar (green), acidic (red), and basic residues (blue).

tissues were harvested at 21 dpi (Fig. 2m). In this group vRNA was detected in the nasal turbinates ($\sim 10^5$–$10^7$ genome copies/g) and lungs ($\sim 10^4$–$10^6$ genome copies/g) of all deer mice and in the small intestine of two deer mice ($\sim 10^4$ genome copies/g), indicating persistence of vRNA in organs that to our knowledge has not been reported in other animal models studied thus far.

Lesions in animals infected at $10^5$ TCID$_{50}$ were largely absent from nasal turbinates except for mild neutrophilic infiltration in the submucosa at 4 dpi (Fig. 3a, left panel). Abundant vRNA was primarily detected in epithelial cells at 2 and 4 dpi (Fig. 3a, middle panels) and anti-genomic RNA could be detected occasionally in epithelial cells at 2 dpi only (Fig. 3a, right panels). Histopathologic examination of lung tissue revealed lesions that included perivascular and peribronchiolar infiltrations of histiocytes and neutrophils with occasional multinucleated syncytial cells (Fig. 3b, left panel). Occasional discrete foci of interstitial pneumonia were observed (Fig. 3b, left panel). Abundant vRNA was primarily detected in bronchiolar epithelial cells and occasionally in the interstitium (Fig. 3b, middle panels). Anti-genomic RNA could be detected in individual scattered bronchiolar epithelial cells (Fig. 3b, right panels). Lung lesions were largely absent at 21 dpi although some foci of inflammation with syncytia were

identified, and vRNA was not detected in the lungs by ISH despite qRT-PCR positivity (Supplementary Fig. 2).

Blood biochemistry and hematological parameters were compared for uninfected ($n = 5$) and SARS-CoV-2-infected deer mice ($n = 3$; $10^6$ TCID$_{50}$) (Fig. 4a–e). Both groups showed similar levels of white blood cells; however, infected deer mice had reduced lymphocyte counts and elevated neutrophil counts (Fig. 4a). These data appear to recapitulate the trend in hematologic values observed in COVID-19 patients, where lymphocytopenia with elevated neutrophil counts can occur[37]. One out of three infected deer mice had a dramatically elevated neutrophil-to-lymphocyte ratio (NLR), a clinical metric that has been found to be elevated in human patients during severe COVID-19[38], and the NLR was significantly elevated in all infected deer mice ($P < 0.05$, Mann–Whitney test). The lack of a difference in alanine aminotransferase (ALT) (Fig. 4c), and the reduced values for blood albumin (ALB) (Fig. 4d) and blood urea nitrogen (BUN) (Fig. 4e), when uninfected and infected deer mice were compared suggests that infection did not cause liver and kidney impairment. Additional serum biochemical values were unremarkable (Supplementary Fig. 3). Transcriptional profiling of host cytokine expression in the lungs by real time PCR was carried out 2 dpi (Fig. 4f) and compared with uninfected controls ($n = 6$). TNFα

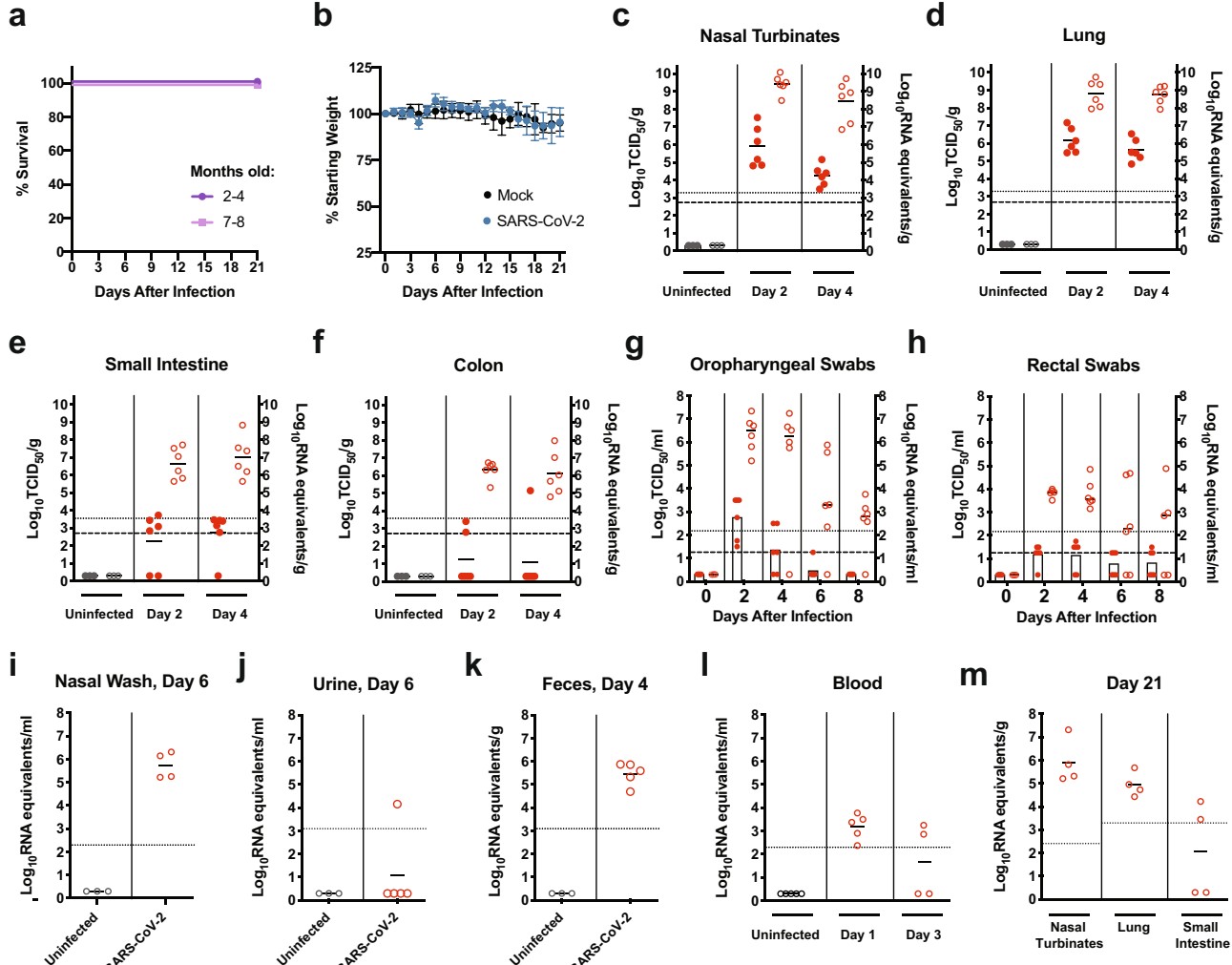

**Fig. 2 SARS-CoV-2 infection of adult deer mice. a–m** Eight to thirty two-week old female or male deer mice (*P. maniculatus*) were inoculated with $10^5$ TCID$_{50}$ or $10^6$ TCID$_{50}$ of SARS-CoV-2 by an intranasal route (i.n.) of administration and compared with age-matched uninfected controls. A summary table outlining the experimental design for the infection studies with reference to the corresponding figure panels is provided (Supplementary Table. 1). Solid lines indicate mean, error bars indicate SD. Dashed lines and dotted lines indicate the limit of detection for the TCID$_{50}$ assay and qRT-PCR assay, respectively. **a, b**, Kaplan–Meier curve depicting survival data (**a**) and weight data (**b**) over the course of 21 days following SARS-CoV-2 exposure (challenge dose, $10^5$ TCID$_{50}$). **c–f**, Infectious viral load (filled in circles, left axis) and vRNA levels (empty circles, right axis) in the (**c**) nasal turbinates, (**d**) lung, (**e**) small intestine, and (**f**) colon (challenge dose, $10^5$ TCID$_{50}$). **g, h**, Infectious viral load (filled in circles with bars, left axis) and viral RNA levels (empty circles, right axis) in (**g**) oropharyngeal swabs and (**h**) rectal swab samples (challenge dose, $10^5$ TCID$_{50}$). **i**, Viral RNA levels in nasal washes at 6 dpi (infectious dose, $10^5$ TCID$_{50}$). The sample was obtained for 4/5 deer mice. **j–l**, Viral RNA levels in the (**j**) urine, (**k**) feces, and (**l**) blood at the indicated times postinfection (infectious dose, $10^6$ TCID$_{50}$). **m**, Viral RNA levels in the nasal turbinates, lung, and small intestines at 21 days postinfection (infectious dose, $10^5$ TCID$_{50}$). Data were collected from two independent experiments. (**a, b**, $n = 6$; **c–f**, $n = 3/6$, for uninfected/infected; **g, h**, $n = 5/6$; **i–k**, $n = 3/4$ (**i**) or 3/5 (**j, k**), for uninfected/infected; **l**, $n = 5$ (uninfected, 1 dpi) or 4 (3 dpi); **m**, $n = 4$ biologically independent samples (individual animals) were examined).

and IL-6 mRNA expression was elevated in all infected mice (mean fold changes of $2^{2.7}$ and $2^{3.5}$, respectively). IL-10 mRNA expression was elevated in five out of six infected animals and reduced in one (mean fold change of $2^{1.5}$). In contrast, IFNα was elevated in two and reduced in four infected deer mice (mean fold change of $2^{-0.8}$). Overall, the mRNA levels of inflammatory cytokines (TNFα, IL-6, and IL-10) were slightly elevated compared with what is seen in the lungs of deer mice (~2–5-fold increase) following experimental infection with SNV[39]. All five deer mice infected at $10^5$ TCID$_{50}$ had detectable serum IgG titers against mixed spike/nucleoprotein (S/N) antigen as assessed by ELISA at 14 dpi (OD 2.7–2.8 at 1:100) ($P < 0.0001$, Student $t$ test) (Fig. 4g), and neutralizing antibodies by 28 dpi (plaque reduction

neutralization test (PRNT$_{90}$) 1:40–1:320, $P = 0.0349$, Student $t$ test) (Fig. 4h).

Transmission by direct contact was examined between SARS-CoV-2-infected and naïve deer mice (experimental design schematic, Supplementary Fig. 4). Five adult male and female deer mice, infected i.n. with $10^6$ TCID$_{50}$ SARS-CoV-2 were each transferred to a new cage at 1 dpi and co-housed with a single naive contact deer mouse (1:1 ratio; zero days post-direct contact (dpc)). Oropharyngeal and rectal swab samples taken every other day revealed that initially one contact animal (DM7, Fig. 5a b) rapidly became infected and shed moderate levels of vRNA as early as 2 dpc. The level of vRNA in both swab samples from DM7 continued to increase for the next 4 (oropharyngeal swabs)

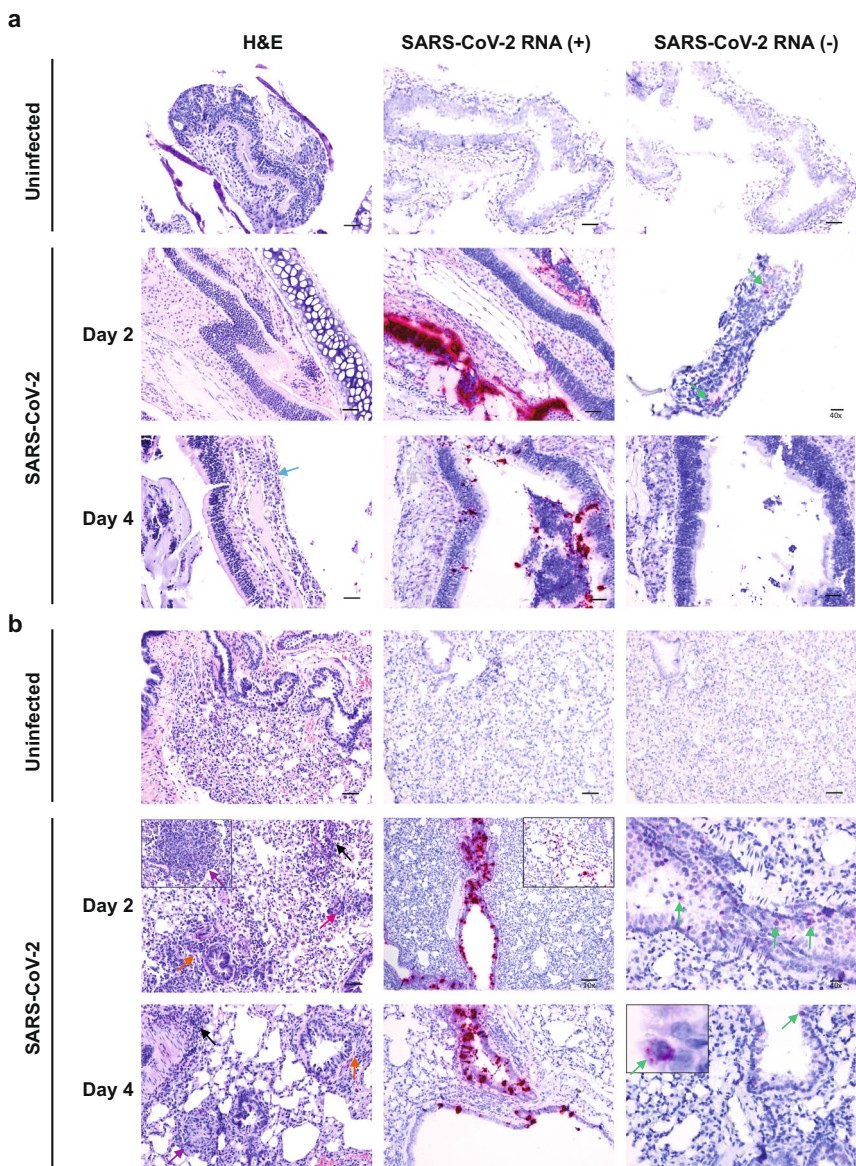

**Fig. 3 Histopathology and virus distribution.** Hematoxylin/eosin (H&E) staining (left) and in situ hybridization (ISH) using antisense probes that detect the SARS-CoV-2 genome/mRNA (middle) and sense probes that detect anti-genomic RNA (right) were carried out on (**a**) nasal turbinates and (**b**) lung tissue of uninfected and SARS-CoV-2-infected deer mice ($10^5$ TCID$_{50}$ i.n. route) at 2 and 4 dpi. Positive detection of viral genomic RNA/mRNA or anti-genomic RNA is indicated by magenta staining (middle/right panels and insets). Arrows indicate perivascular infiltrations of histiocytes and neutrophils (black), peribronchiolar infiltrations of histiocytes and neutrophils (orange), mild neutrophilic infiltration in the submucosa (blue), occasionally observed discrete foci of interstitial pneumonia (purple), occasionally observed multinucleated syncytial cells (magenta), and anti-genomic RNA occasionally in individual scattered bronchiolar epithelial cells (green and right inset). The magnification is ×20 for H&E and ×20 for ISH unless otherwise indicated. Scale bars =100 μm, 50 μm, and 20 μm for 10×, 20×, and 40×, respectively. A total of three biological replicates (individual animals) were assessed at each time point, and the selected panels are representative of these findings.

or 2 (rectal swabs) days and then declined. The remaining contact deer mice had detectable vRNA in oropharyngeal and rectal swab samples at low levels from 2 to 4 dpc which declined until 10 dpc when shed vRNA was again detected in both oropharyngeal and rectal swabs from all contact deer mice (including DM7). Interestingly, despite the high challenge dose administered to donor deer mice only low amounts of infectious virus were detected in swabs (Fig. 5c d). Infectious virus was detected in oropharyngeal and rectal swabs of contact deer mouse DM7 at 4 dpc and an oropharyngeal swab sample from contact deer mouse DM6 at 10 dpc. An additional transmission study was carried out in the same manner and nasal wash, nasal turbinates, and lung samples were collected from contact animals at 2 and 4 dpc (Fig. 5e f). Viral

RNA was detected in nasal wash, nasal turbinates, and the lungs of contact deer mice DM19 (2 dpc) and DM30 (4 dpc), as well as in the nasal wash of deer mouse DM17 (2 dpc), DM26 (4 dpc) and DM28 (4 dpc). IgG against mixed SARS-CoV-2 spike/ nucleoprotein antigens were detected in deer mouse DM7 at 21 dpc, indicating seroconversion and confirming that direct contact transmission had occurred (Fig. 5g). A similar, but more stringent transmission study was carried out using a lower dose of virus ($10^5$ TCID$_{50}$), and donor and naive deer mice were mixed at 2 dpi (Fig. 5h, i). Oropharyngeal and rectal swab samples taken every other day revealed that contact deer mice had detectable vRNA in both oropharyngeal and rectal swab samples from 2 to 4 dpc at low levels, which declined until 6–10 dpc after which shed vRNA

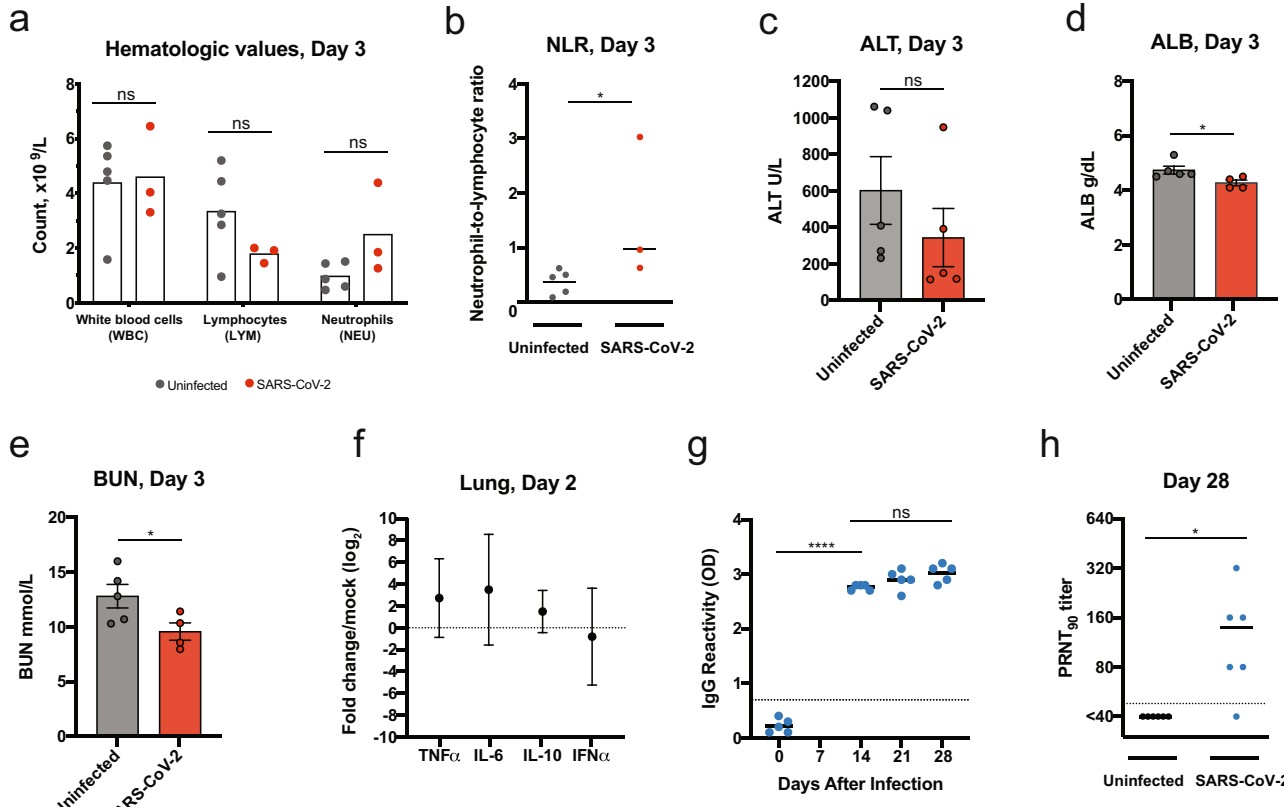

**Fig. 4 Deer mouse host response to SARS-CoV-2 infection. a–e** Hematological levels and serum biochemistry were measured in uninfected and SARS-CoV-2-infected deer mice (infectious dose $10^6$ TCID$_{50}$, i.n. route) at 3 dpi, including (**a**) white blood cell, lymphocyte, and neutrophil counts, (**b**) the neutrophil-to-lymphocyte ratio, (**c**) alanine aminotransferase (ALT), (**d**) blood albumin (ALB), and (**e**) blood urea nitrogen (BUN). **f** Cytokine gene expression was measured for TNFα, IL-6, IL-10, and IFNα in the lungs of SARS-CoV-2-infected deer mice and displayed relative to age-matched mock-infected animals. Gene expression was normalized using GAPDH as a control. **g** IgG antibody response against SARS-CoV-2 mixed spike and nucleoprotein (S/N) antigens were assessed by ELISA using serum collected on the indicated days postinfection (infectious dose, $10^5$ TCID$_{50}$). **h** Neutralizing antibody against SARS-CoV-2 was measured by PRNT$_{90}$ using serum collected at 28 days after SARS-CoV-2 infection (infectious dose, $10^5$ TCID$_{50}$). Dotted lines indicate the limit of detection. Solid lines and bars indicate mean, error bars indicate SEM (**c–e**) or 95% confidence interval (**f**). Data were collected from two independent experiments (**a**, **b**, $n = 5/3$ for uninfected/infected; **c**, $n = 5$; **d**, $n = 5$, **e**, $n = 5/4$ for uninfected/infected, **f**, $n = 6$, **g**, $n = 5$; **h**, $n = 6$ biologically independent samples (individual animals) were examined). $*P < 0.05$, $****P < 0.0001$, ns $= P > 0.05$; two-sided, unpaired Student $t$ test (**a–e**, **g–h**), $P = 0.0357$ (**b**), 0.0348 (**d**), 0.0474 (**e**), <0.0001 (**g**), and 0.0349 (**h**), Mann–Whitney test (**b**).

was again detected in oropharyngeal swabs from four out of five and rectal swabs from three out of five contacts. At 21 dpc vRNA was detected in the oropharyngeal swabs of two of five contact and was not detected in any rectal swab samples.

## Discussion

The described experiments demonstrate that adult deer mice are susceptible to experimental infection with a human isolate of SARS-CoV-2, resulting in asymptomatic infection or mild disease with lesions limited to mild lung pathology, despite a high viral burden and elevated levels of inflammatory cytokines in the lungs and seroconversion. The detected RNA viremia in SARS-CoV-2-infected deer mice appears to be more transient than what is observed in deer mice experimentally infected with SNV[40]; however, the viral shedding and the propensity for direct contact transmission appears to be greater[35,40]. The relatively low amount of shed infectious virus from nasal, oral, and rectal routes consistently resulted in direct contact transmission from infected to co-housed naive deer mice. SNV is also detectable in deer mouse feces and urine, and zoonotic transmission of SNV into humans is thought to result from unintentional ingestion or inhalation of contaminated deer mouse excreta[41]. Additional studies should be

carried out to determine if vRNA detected in the feces and occasionally urine is indicative of the presence of infectious virus.

Concerns have justifiably been raised about the potential for reverse zoonosis, human-to-animal transmission of SARS-CoV-2 into susceptible wild animal host species[31]. The introduction of novel pathogens into susceptible wildlife hosts can have devastating effects on wildlife populations[42]. The findings reported here suggest that the impacts of SARS-CoV-2 on infected *Peromyscus* species rodents are likely to be minimal; however, wild deer mouse populations may be more or less susceptible to infection than the experimentally housed animals we have described. An additional concern is the potential for zoonotic maintenance of SARS-CoV-2 in an animal reservoir and/or geographic region where SARS-like coronaviruses had not previously been endemic. The findings reported here are concerning in light of the fact that *Peromyscus* species rodents tolerate persistent infection with and serve as the primary reservoirs for several emerging zoonotic pathogens that spillover into humans, including *Borrelia burgdorferi*[32], DTV[33], and SNV[34,35]. It should be acknowledged that although deer mice are widely distributed in North America and can live in close proximity to humans, the actual risk of human-to-deer mouse transmission remains unknown. Further, it remains unclear how well the transmission

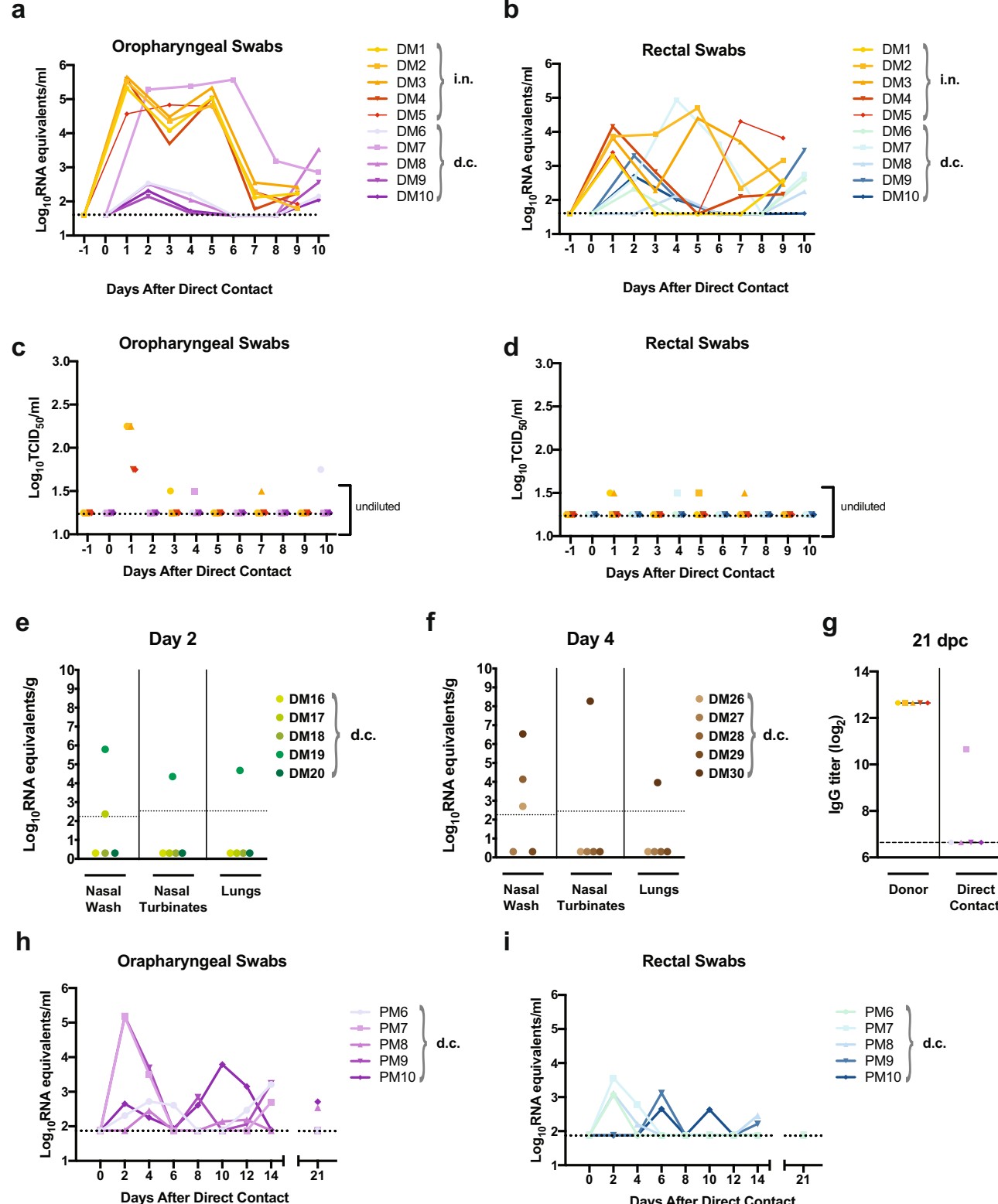

studies we describe translate to the risk of sustained transmission within wild deer mouse populations. The theoretical risk of zoonotic transmission of SARS-CoV-2 back to humans from *Peromyscus* species rodents would depend on additional unknowns such as whether infectious virus is present in excreta at high enough levels to initiate human infection and whether other wild or domestic animal species might serve as an intermediary host. Efforts to monitor wild, especially peri-domestic *Peromyscus*

rodent populations are likely warranted as the SARS-CoV-2 pandemic continues to progress.

Last, we suggest that the deer mouse model of SARS-CoV-2 infection, as a largely outbred small animal model, may prove useful for studying viral pathogenesis, particularly the determinants of asymptomatic infection, viral persistence, and transmission, and for evaluating the protective efficacy of experimental vaccines and therapeutics.

**Fig. 5 Transmission of SARS-CoV-2 between deer mice through direct contact. a–g** Adult male and female deer mice were exposed to $10^6$ TCID$_{50}$ SARS-CoV-2 by an i.n. route of infection (a schematic of the experimental design is provided in Supplementary Fig. 4). At 1 dpi individual inoculated donor deer mice were transferred to a new cage and co-housed with a single naïve deer mouse (1:1 ratio) to assess SARS-CoV-2 transmission by direct contact (d.c.). Deer mice were either maintained in direct contact throughout the 10 day study for serial swabbing or humanely euthanized on 2 or 4 days post-contact (dpc) for tissue collection. **a–d**, Viral RNA levels (**a, b**) and infectious viral loads (**c, d**) were measured in oropharyngeal swabs and rectal swab samples every other day from -1 to 10 days after direct contact was initiated. Oropharyngeal or rectal swab samples obtained from the same animal are demarcated with a unique color, and samples derived from co-housed pairs share the same symbol. **e, f,** An additional transmission study was carried out in the same manner and nasal wash, nasal turbinates, and lung samples were collected from contact animals at (**e**) 2 dpc and (**f**) 4 dpc. **g,** IgG antibody response (reciprocal serum dilution) against SARS-CoV-2 S and N in donor and direct-contact exposed animals was assessed by ELISA using serum collected at 21 dpc. **h, i** In an additional study adult male and female deer mice were exposed to $10^5$ TCID$_{50}$ SARS-CoV-2 by an intranasal route of infection. At 2 dpi individual inoculated donor deer mice were transferred to a new cage and co-housed with a single naïve deer mouse (1:1 ratio) to assess SARS-CoV-2 transmission by direct contact ($n = 5$). Viral RNA levels were measured in (**h**) oropharyngeal swabs and (**i**) rectal swab samples every other day from 0 to 14 days after direct contact was initiated, and the final swab samples were collected at 21 dpi and assessed. Dotted or dashed lines indicate the limit of detection. Solid lines indicate means. Data were collected from two independent experiments.

## Methods

**Ethics statement.** The experiments described in this study were carried out at the National Microbiology Laboratory (NML) at the Public Health Agency of Canada as described in the Animal use document AUD# H-20-006. Experiments were approved by the Animal Care Committee located at the Canadian Science Center for Human and Animal Health in accordance with the guidelines provided by the Canadian Council on Animal Care. All procedures were performed under anesthesia, and all efforts were made to minimize animal suffering and to reduce the number of animals used. All infectious work was performed under biosafety level 3 (BSL-3) conditions or higher.

**Viruses.** The SARS-CoV-2 strain used in these studies (SARS-CoV-2; hCoV-19/Canada/ON-VIDO-01/2020, GISAID accession# EPI_ISL_425177) was isolated from a clinical specimen obtained at the Sunnybrook Research Institute (SRI)/University of Toronto on VeroE6 cells and provided by the Vaccine and Infectious Disease Organization (VIDO) with permission. The P1 virus was subsequently passaged at a 1:1000 dilution on mycoplasma-free VeroE6 cells (ATCC) in Dulbecco's Modified Eagle's Medium (Hyclone) containing 1% L-glutamine and 0.5 µg/ml of TPCK-trypsin and harvested when 80% cytopathic effect (CPE) became evident. The P2 virus stock was clarified by centrifugation at $6000 \times g$ for 5 min and stored at $-80\,°C$ until thawed for deer mouse infections. The virus stock was titrated on Vero cells by conventional TCID$_{50}$ assay, as described previously[44].

**Deer mouse challenge experiments.** Deer mice (*Peromyscus maniculatus rufinus*) used in these studies were supplied by a breeding colony housed at the University of Manitoba in a pathogen-free facility. All the deer mice were acclimated for a minimum of one week prior to the initiation of experimental procedures. Deer mice were randomly assigned to their respective groups and were housed in a temperature-controlled, light-cycled facility. Deer mice were supplied with food and water ad libitum and were monitored daily throughout the course of the experiments. Thirteen to thirty two-week old male or female deer mice were infected with $10^5$ or $10^6$ TCID$_{50}$ of SARS-CoV-2 by an intranasal route (i.n.) of administration in a 50-µl volume. Blinding of the animal experiments was not performed. Sample sizes were calculated a priori, and no animals were excluded from the data analysis although at times sample volumes were insufficient to allow downstream analyses.

**Blood, fluids, feces, swab, and tissue collection.** All deer mice were exsanguinated via cardiac puncture under deep isoflurane anesthesia prior to euthanization. Whole blood and serum was collected in BD microtainer tubes (K$_2$-EDTA or serum, respectively) (Becton, Dickinson and Company) as per the manufacturer's instructions. Serum samples were collected from the retro-orbital sinus under deep isoflurane sedation at 14, 21, and 28 dpi for challenge studies and at 21 dpc for transmission studies. Oropharyngeal and rectal swabs were taken using fine tip Dryswab Fine Tip rayon swabs (MWE cat #MW113). Harvested tissue samples for infectious assays were flash-frozen and stored at $-80\,°C$ until later use, and swab samples were placed in 1 ml of MEM supplemented with 1% heat-inactivated FBS, 1x L-glutamine, and 2x penicillin–streptomycin, flash frozen, and stored at $-80\,°C$ until later use. Tissue samples harvested for vRNA detection were immersed in RNAlater (Ambion) at $4\,°C$ for 1 day, then stored at $-80\,°C$ until later use. Nasal washes were obtained at necropsy by introducing 500 µl of PBS with 2x penicillin–streptomycin into the nasal cavity through an incision in the exposed trachea and collecting the drained fluid from the nares into a petri dish. For urine collection deer mice were gently raised from their cages using soft-tip mouse handling forceps and placed immediately onto a 35-mm petri dish, a process that consistently allowed a minimum of 25 µl of urine to be collected. Feces samples were collected fresh from the isoflurane chamber following sedation, homogenized in 0.5 ml of 0.89% NaCl per 80 mg of feces using a Bead Ruptor Elite Bead Mill Homogenizer (Omni International) with a stainless steel bead at 4 m/s for 30 s. The

sample was then clarified by centrifugation for 20 min at $4000 \times g$ and the supernatant was filtered through a 0.2-µm filter. Viral RNA was then isolated using the QIAamp Viral RNA Mini kit (Qiagen), as per the manufacturer's instructions.

**Infectious virus in tissues and swab samples.** For infectious virus assays, thawed tissue samples were weighed and placed in 1 ml of MEM supplemented with 1% heat-inactivated FBS, 1x L-glutamine before being homogenized in a Bead Ruptor Elite Bead Mill Homogenizer (Omni International) at 4 m/s for 30 s and clarified by centrifugation at $1500 \times g$ for 6 min. Swab samples were vortexed for 10 s and clarified at $540 \times g$ for 5 min. Tissue homogenates or swab samples were serially diluted 10-fold in MEM supplemented with 1% heat-inactivated FBS, 1x L-glutamine, and 2x penicillin–streptomycin. One-hundred microliter volumes of the samples (lowest dilution was 1:10 and neat for tissues and swab samples, respectively) were added to 96-well plates of 95% confluent Vero cells containing 50 µl of the same medium in replicates of three and incubated for 5 days at $37\,°C$ with 5% $CO_2$. Plates were scored for the presence of cytopathic effect on day 5 after infection.

**Viral RNA copies in tissues, fluids, and swab samples.** For vRNA copy number analysis, tissue samples were thawed and weighed and homogenized in 600 µl of RLT buffer using a Bead Ruptor Elite Bead Mill Homogenizer (Omni International) with a stainless steel bead at 4 m/s for 30 s. Viral RNA from 30 mg tissue samples was extracted with the RNeasy Plus Mini kit (Qiagen), and vRNA was extracted from swab samples (140 µl) and fluids (nasal wash (140 µl), urine (25 µl), feces (80 mg), and blood (140 µl)) with the QIAamp Viral RNA Mini kit (Qiagen). A SARS-CoV-2 E-specific real time RT–PCR assay was used for the detection of vRNA[45]. RNA was reverse transcribed and amplified using the primers E_Sarbeco_F1, E_Sarbeco_R2, and probe E_Sarbeco_P1 (listed in Supplementary Table 2) using the TaqPath 1-Step Multiplex Master Mix kit (Applied Biosystems) on a QuantStudio 5 real time PCR system (Appiled Biosystems), as per the manufacturer's instructions. A standard curve was generated in parallel for each plate using synthesized DNA and used for the quantification of viral genome copy numbers. The Design and Analysis Software version 1.5.1 (ThermoFisher Scientific) was used to calculate the cycle threshold values, and a cycle threshold value $\leq36$ for both replicates was considered positive.

**Blood counts and biochemistry.** Complete blood counts were carried out using a VetScan HM5 hematology system (Abaxis Veterinary Diagnostics), as per the manufacturer's instructions. Analysis of serum biochemistry was performed with a VetScan VS2 analyzer (Abaxis Veterinary Diagnostics), as per the manufacturer's instructions.

**Transcriptional profiling of host responses.** RNA was extracted from tissues using the RNeasy plus mini kit (Qiagen) following the manufacturer's instructions. A two-step qRT-PCR reaction in triplicate was performed on a Quantstudio 5 (Applied Biosystems). Reverse transcription was carried out with the Superscript III RT first strand synthesis kit (Invitrogen) following DNase elimination of genomic DNA. The RT reaction was performed using 100 ng of template RNA mixed with random hexamers as primers in a 20 µL reaction volume for 5 min at $65\,°C$ followed by 10 min at $25\,°C$ and 50 min at $50\,°C$. The qPCR reaction was carried out using the PowerUp SYBR Green Master Mix (Applied Biosystems) with 2 µL of cDNA. The cycle parameters for qPCR were 2 min at $50\,°C$ and 2 min at $95\,°C$ followed by 40 cycles of 3 s at $95\,°C$ and 30 s at $60\,°C$. The qPCR reactions were performed in a 20 µL volume with oligonucleotide pairs specific for deer mouse TNFα, IL6, IL10, and IFNα[46] at a concentration of 2 µM. The fold change in gene expression of TNFα, IL6, IL10, and IFNα in SARS-CoV-2-infected versus uninfected deer mice was calculated using the ΔΔCt method with GAPDH as a

reference gene[47]. GraphPad Prism's multiple t test was used to perform the second subtraction so as not to lose the variation of the mock animals.

**SARS-CoV-2-S-specific enzyme-linked immunosorbent assay (ELISA).** SARS-CoV-2 spike/nucleoprotein (S/N)-specific IgG antibody responses were assessed using an in-house assay. Briefly, a 1:100 dilution (or a serial dilution) of deer mouse serum was carried out in duplicate and added to plates pre-coated with both the spike and nucleoprotein (S/N) antigens in the same assay wells. Deer mouse IgG was detected with a KPL peroxidase-labeled polyclonal goat antibodies against *Peromyscus leucopus* IgG (H + L) (Sera Care).

**Virus neutralization assay.** Deer mouse serum samples were collected and stored at −80 °C. SARS-CoV-2 stocks were titrated and used in the plaque reduction neutralization test (PRNT)[48]. Briefly, serum was heat-inactivated at 56 °C for 30 min and diluted 2-fold from 1:40 to 1:1280 in DMEM supplemented with 2% FBS. Diluted sera were incubated with 50 plaque forming units of SARS-CoV-2 at 37 °C and 5% $CO_2$ for 1 h. The sera-virus mixtures were added to 24-well plates containing Vero E6 cells at 100% confluence, followed by incubation at 37 °C and 5% CO2 for 1 h. After adsorption, 1.5% carboxymethylcellulose diluted in MEM supplemented with 4% FBS, L-glutamine, nonessential amino acids, and sodium bicarbonate was added to each well and plates were incubated at 37 °C and 5% CO2 for 72 h. The liquid overlay was removed and cells were fixed with 10% neutral-buffered formalin for 1 h at room temperature. The monolayers were stained with 0.5% crystal violet for 10 min and washed with 20% ethanol. Plaques were enumerated and compared with a 90% neutralization control. The PRNT$_{90}$ end-point titers was defined as the highest serum dilution resulting in 90% reduction in plaques. PRNT$_{90}$ titers ≥1:40 were considered positive for neutralizing antibodies.

**Histopathology and vRNA in situ hybridization.** Tissues were fixed in 10% neutral phosphate buffered formalin for two to four weeks. Subsequently, routine processing was carried out and tissue samples were sectioned at 5 μm. A set of slides was stained with hematoxylin and eosin for histopathologic examination. RNA in situ hybridization (ISH) was carried out using RNAscope 2.5 HD Detection Reagent-Red (Advanced Cell Diagnostics), according to the manufacturer's instructions. Briefly, formalin-fixed, paraffin-embedded tissue samples of the various tissues were mounted on slides, baked in a dry oven for 1 h at 60 °C, and deparaffinized. Tissue sections were then pre-treated with RNAscope $H_2O_2$ to block endogenous peroxidases for 10 min at room temperature. Target retrieval was carried out using the RNAscope Target Retrieval Reagent for 15 min. RNA-scope Protease Plus Reagent was then applied for 15 min at 40 °C. The probes targeting SARS-CoV-2 RNA (V-nCoV2019-S probe, ref#848561) or anti-genomic RNA (V-nCoV2019-S-sense ref#845701) were designed and manufactured by Advanced Cell Diagnostics. The negative probe was also obtained from Advanced Cell Diagnostics (Reference # 310034). The stained tissues were counterstained with Gills I Hematoxylin, and the final images were captured using a light microscope equipped with a digital camera.

**SARS-CoV-2 transmission studies.** Eight to thirty two-week old female or male deer mice were challenged with $10^5$ or $10^6$ TCID$_{50}$ of SARS-CoV-2 by an intranasal route (i.n.) of administration in a 50 μl volume. At 24 or 48 hpi (for $10^6$ TCID$_{50}$ and $10^5$ TCID$_{50}$, respectively) experimentally infected deer mice were placed into a fresh cage with a naïve cage mate to assess direct contact transmission (donor: contacts at 1:1 ratio). Deer mice were monitored daily for clinical signs and/or weight loss. Oropharyngeal and rectal swab samples were taken on alternate days up to 10 DPI or 14 DPI (for $10^6$ TCID$_{50}$ and $10^5$ TCID$_{50}$, respectively), and oropharyngeal and rectal swabs as well as serum samples were collected at 21 days post-contact (dpc).

**Bioinformatic analyses.** ACE2 sequences were aligned with Clustal Omega[49]. For the phylogenetic analysis, mitochondrial genome sequences (see Accession codes) were aligned with MAFFT v7.467[50], with regions of poor alignment trimmed with Gblocks v0.91b[51] resulting in a final alignment of 15,393 bp in 115 blocks of minimum 5 bp lengths. A maximum likelihood phylogeny was constructed with RAxML v8.2.12[52] using the GTR + I + G4 substitution model as selected by modeltest-ng[53].

**Data analysis.** The results were analyzed and graphed using Prism 8 software (Graphpad Software). As appropriate, statistical analyses were performed using ANOVA with multiple comparison correction, the multiple *t* test, or the unpaired *t* test with Welch's correction or Mann–Whitney test.

**Accession codes.** ACE2 sequences: Deer Mouse (*Peromyscus maniculatus bairdii*; XP_006973269), White-footed Mouse (*Peromyscus leucopus*; XP_028743609), Syrian Hamster (*Mesocricetus auratus*; XP_005074266), Chinese Hamster (*Cricetulus griseus*; XP_003503283), Mouse (*Mus Musculus*; ABN80105), Human (*Homo sapiens*; ACT66268), Masked Palm Civet (*Paguma larvata*; AAX63775), Fruit Bat (*Rousettus aegyptiacus*; XP_015974412), Chinese rufous horseshoe bat (*Rhinolophus sinicus*; ADN93472.1), Domestic cat (*Felis catus*; AAX59005), Ferret (*Mustela*

putorius furo; BAE53380), and Prairie Vole (*Microtus ochrogaster*; XP_005358818). Mitochondrial genomes: *Peromyscus maniculatus* (NC_039921.1), *Peromyscus leucopus* (NC_037180.1), *Mesocricetus auratus* (NC_013276.1), *Cricetulus griseus* (NC_007936.1), *Mus musculus* (NC_005089.1).

**Reporting summary.** Further information on research design is available in the Nature Research Reporting Summary linked to this article.

## Data availability

All of the data supporting the findings of this work can be found within the paper and the accompanying Supplementary Source Data file. The raw data are available from the corresponding author upon request or in the corresponding source data file that is provided with this paper. Source data are provided with this paper.

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

## Acknowledgements

We thank Samira Mubareka of Sunnybrook Research Institute and University of Toronto for providing the clinical sample and Darryl Falzarano of Vaccine and Infectious Disease Organization - International Vaccine Centre (VIDO-InterVac) for isolation of the SARS-CoV-2 strain used in these studies. The authors also thank Michelle French, Kimberly Azaransky, Stephanie Kucas and Christine DeGraff of the Veterinary Technical Services at the National Microbiology Laboratory (NML) for their technical assistance during the course of this work. The work described in this paper was financially supported by the Public Health Agency of Canada and the Canadian Food Inspection Agency.

## Author contributions

The experiments were conceived of and designed by B.D.G., D.S., and D.K. The phylogenic analysis was performed by A.T.D. The ACE2 alignment analysis was carried out by B.D.G. and E.M. The virus was propagated and titered by A.L., and D.K. Animal husbandry, infection and sample collection was carried out by B.D.G., M.C., N.T., B.M.W., A.A., K.T., and D.K. Infectious virus assays were performed by B.D.G. and M.C. Sample inactivation and RNA extractions were carried out by B.D.G., N.T., A.L., B.M.W., L.G., K.N.T., R.V., and ABo. Real-time RT–PCR was performed by B.D.G. and N.T. Transcriptional profiling of host responses was carried out by B.M.W. and N.T. Blood counts, biochemistry, and pathology analysis was performed and interpreted by B.D.G., A.B., G.P., J.S., C.E.H., and D.K. ELISA was developed and carried out by A.L. and S.H. PRNT assays were developed and carried out by E.M., H.W., and M.D. Histopathology and ISH was carried out by EMo and pathology was assessed by CEH. Statistical analysis was performed by B.D.G. and J.A. Figures were prepared by B.D.G., E.M., and L.B. Interpretation of potential impacts on wild deer mice and deer mouse ecology was done by L.R.L. and C.M.J. The initial draft of the paper was written by B.D.G. and D.K., with all other authors contributing to editing into the paper into its final form. The work was managed and supervised by B.D.G., M.C., L.F., C.E.H., and D.K.

## Competing interests

The authors declare no conflicts of interest.
