## [Peer Review File · Nature Communications]

REVIEWER COMMENTS

Reviewer #1 (Remarks to the Author):

The manuscript entitled “SARS-CoV-2 infection and transmission in the North American deer mouse” by Griffin et al. investigates the susceptibility of deer mice to SARS-CoV-2. The authors report that intranasal exposure to SARS-CoV-2 results in replication in the respiratory tract with shedding detected in nasal, oral, and rectal swabs. However, no disease was noted in any of the animals. The authors also show that infected deer mice can transmit SARS-CoV-2 to naïve animals by direct contact. Importantly, the study points out the risk of reverse zoonosis and raises the possibility that SARS-CoV-2 has the potential to become established in deer mice which could serve as a reservoir thus having important public health implications. Overall, this is a really nice study from an established group.

Comments:

1. The experimental design for the various studies performed in lines 112-181, i.e., all of the initial infection studies is very difficult to follow. It would help to have a Table or schematic Figure even if as Extended Data showing the various groups, sizes of groups, and challenge doses etc.
2. Figure 2. Are the authors sure that any cytopathic effect detected in small intestine and colon were caused by SARS-CoV-2 and not contaminating bacteria? It is unusual to see infectious virus assays performed on gastrointestinal tissues.

Reviewer #2 (Remarks to the Author):

In this manuscript, the authors address the question of whether SARS-CoV-2 has the potential of “reverse zoonosis,” and whether it can be transmitted from humans to wildlife. In other words, can new host reservoirs of SARS-CoV-2 become established after spillover from humans? The studies focused on experimentally infected deer mice, one of the most abundant rodents in North America. Deer mice are natural hosts for a number of human pathogens that cause severe disease, including Lyme disease and hantavirus pulmonary syndrome. The authors tested not only the susceptibility of deer mice to SARS-CoV-2, but also the possibility of its transmission to naïve animals.

The manuscript is well written and the conclusions are mostly well justified. I have few minor suggestions for improvement.

Figure 2. Please explain why the limits of detection fluctuate between the samples and swabs, c-f versus g-h. Also, in panel I, results from only 4 of the 5 infected animals are reported.

Figure 3. This figure contains a lot of information and as a result is a little hard to follow. Some of the arrows are difficult to see and the claims shown are hard to justify. For example, do the green arrows point to epithelial cells in the insert of the last panel? What are the blue arrows indicating?

Figure 4. Why is the analysis of analytes based on a higher infectious dose?

Figure 5. I think it would be useful to have the experimental schematic design extended from Figure 4 to the top of Figure 5. Again, what is the rationale for using the different TCID₅₀s? Also, please define d.c. in the figure legend.

Reviewer #3 (Remarks to the Author):

This is the first report of SARS-CoV-2 infection and transmission in North American deer mouse. Because the infection is asymptomatic or causes very mild symptoms, it cannot be used as the model for symptomatic human COVID-19. The significance and value of the work are therefore compromised. In addition to reverse zoonosis mentioned in the paper, whether deer mice might serve as a model for asymptomatic infection in humans merits more exploration and discussion. For example, whether and how the immune response to SARS-CoV-2 could be studied in deer mice should be determined and discussed. Some specific concerns are listed below:

- 1) The animal model is relatively artificial, using $10^5 \sim 10^6$ TCID₅₀ for virus inoculation.
- 2) Since the age of the deer mice used ranges from 8 to 32 weeks old, there will be huge difference in terms of initial body weight and immune response.
- 3) The number, age, and gender of deer mice should be described clearly in all experiments.
- 4) Fig 3: The resolution of the figure is too low.
- 5) Fig 4: Statistical analysis is not sufficiently robust.
- 6) Fig 5: The colors for different samples are quite indistinguishable (Fig 5e, 5f and 5h). The plots for samples are also too small to be seen (Fig 5c & 5d).

Reply to Reviewer Comments

Manuscript: SARS-CoV-2 infection and transmission in the North American deer mouse

We thank the reviewers very much for their constructive comments. The manuscript has certainly benefited by addressing these suggestions and comments, and we hope that our reply to the comments has satisfactorily addressed these concerns.

Reviewer #1 (Remarks to the Author):

The manuscript entitled “SARS-CoV-2 infection and transmission in the North American deer mouse” by Griffin et al. investigates the susceptibility of deer mice to SARS-CoV-2. The authors report that intranasal exposure to SARS-CoV-2 results in replication in the respiratory tract with shedding detected in nasal, oral, and rectal swabs. However, no disease was noted in any of the animals. The authors also show that infected deer mice can transmit SARS-CoV-2 to naïve animals by direct contact. Importantly, the study points out the risk of reverse zoonosis and raises the possibility that SARS-CoV-2 has the potential to become established in deer mice which could serve as a reservoir thus having important public health implications. Overall, this is a really nice study from an established group.

Comments:

1. The experimental design for the various studies performed in lines 112-181, i.e., all of the initial infection studies is very difficult to follow. It would help to have a Table or schematic Figure even if as Extended Data showing the various groups, sizes of groups, and challenge doses etc.

Thank you for identifying this shortcoming. Following the suggestion of the reviewer we have added an Extended Data file outlining the various groups and the experimental design for each experiment (group, group size, challenge dose). The text was also modified as follows:

Line 115: “Here, eight to thirty two-week old male or female deer mice (*P. maniculatus rufinus*; in-house colony) were inoculated with 10^5 TCID₅₀ of SARS-CoV-2 by an intranasal route (i.n.) and monitored daily for clinical signs and weight loss for 21 days or necropsied at 2 and 4 day post-infection (dpi). **A table outlining the experimental design is provided in Extended Data Fig. 5).**

Line 135: “An additional experimental group of deer mice (n = 5) were inoculated i.n. with 10^6 TCID₅₀ of SARS-CoV-2. **A schematic depicting the experimental design is provided in Extended Data Fig. 5)**

Line 365: “**a - m** Eight to thirty two-week old female or male deer mice (*P. maniculatus*) were inoculated with 10^5 TCID₅₀ or 10^6 TCID₅₀ of SARS-CoV-2 by an intranasal route (i.n.) of

administration and compared to age-matched uninfected controls. A summary table outlining the experimental design for the infection studies with reference to the corresponding figure panels is provided (Extended Data Fig. 5).”

2. Figure 2. Are the authors sure that any cytopathic effect detected in small intestine and colon were caused by SARS-CoV-2 and not contaminating bacteria? It is unusual to see infectious virus assays performed on gastrointestinal tissues.

This is an interesting point that we have considered as well, and we have experience with infectious virus assays for other viruses, such as some influenza viruses, that replicate in the intestinal tract. We have determined from previous influenza work that a combination of prior removal of the feces from the lumen of the digestive tract, homogenization of the tissues with 2x the typical dose of antibiotics (2 x pen/strep), and centrifugation at 1500 x g for 6 minutes eliminates observable effects of bacterial contamination during the timespan of the assay. We observed no similar cytopathic effect (CPE) in digestive tract tissues from the uninfected deer mice. We also routinely assess infectious virus in rectal swabs using 2x pen/strep, and we note that in the first transmission study DM7 was the sole deer mouse with detectible infectious virus in a rectal swab sample, and this same animal also had the highest level of viral RNA in the same swab sample.

Reviewer #2 (Remarks to the Author):

In this manuscript, the authors address the question of whether SARS-CoV-2 has the potential of “reverse zoonosis,” and whether it can be transmitted from humans to wildlife. In other words, can new host reservoirs of SARS-CoV-2 become established after spillover from humans? The studies focused on experimentally infected deer mice, one of the most abundant rodents in North America. Deer mice are natural hosts for a number of human pathogens that cause severe disease, including Lyme disease and hantavirus pulmonary syndrome. The authors tested not only the susceptibility of deer mice to SARS-CoV-2, but also the possibility of its transmission to naïve animals. The manuscript is well written and the conclusions are mostly well justified. I have few minor suggestions for improvement.

Figure 2. Please explain why the limits of detection fluctuate between the samples and swabs, c-f versus g-h.

This is an important point that is dependent on the assay and we thank the reviewer for this question. We have noted some missing information from the methods section that has now been added and added to the text to clarify this point (please, see below). For the quantification of viral RNA (vRNA) in tissues (c-f) we weighed the tissues and extracted RNA from 30 mg of tissue and processed the samples with the RNeasy Plus Mini kit (Qiagen). For the quantification of vRNA in liquid samples and feces we used the QIAamp Viral RNA Mini kit (Qiagen) with a starting volume of 140 µl. This resulted in a different LOD for these two sets of panels. The LOD differs in panels i-l since the sampling volume differed for the various liquid samples, as follows:

Swabs: 1 ml sample volume, 140 µl processed.

Blood: 140 µl of blood processed.

Urine: 25 µl of urine processed.

Feces: 80 mg in 500 µl, 140 µl processed.

We also generated a separate standard curve for each qPCR run using a DNA control and calculated the limit of detection (LOD) by using a ct of 36 cycles that slightly impacted the LOD from run to run.

For cell culture assays to quantify infectious virus, we consistently used a volume of 1 ml for homogenizing tissues and for swab samples. For each sample type we used the infectious titer that would be calculated if 2/3 wells were positive for viral cytopathic effect at the lowest dilution tested in the assay as the LOD; however, the lowest dilution we used for tissues was 1:10 (c-f), whereas we ran neat samples for the swab samples (g-h) to increase the sensitivity. This resulted in the different LODs. We added the following text to clarify:

Line 511: “One hundred microliter volumes of the samples (lowest dilution was 1/10 and neat for tissues and swab samples, respectively) were added to 96-well plates of 95% confluent Vero cells containing 50 µl of the same medium in replicates of three and incubated for 5 days at 37 °C with 5% CO₂. Plates were scored for the presence of cytopathic effect on day 5 after infection.”

We also added the following lines to the text to clarify how the nasal washes were collected and how the fluids were processed:

Line 493: “Nasal washes were obtained at necropsy by introducing 500 µl of PBS with 2x penicillin–streptomycin into the nasal cavity through an incision in the exposed trachea and collecting the draining fluid from the nares into a petri dish.”

Line 515 (heading) and 516: “Viral RNA copies in tissues, fluids, and swab samples
For vRNA copy number analysis, tissue samples were thawed and weighed and homogenized in 600 µl RLT buffer using a Bead Ruptor Elite Bead Mill Homogenizer (Omni International) with a stainless steel bead at 4 m/s for 30 seconds. Viral RNA from 30 mg tissue samples was extracted with the RNeasy Plus Mini kit (Qiagen), and vRNA was extracted from swab samples (140µl) and fluids (nasal wash (140µl), urine (25 µl), feces (80 mg), and blood (140µl)) with the QIAamp Viral RNA Mini kit (Qiagen).”

Also, in panel I, results from only 4 of the 5 infected animals are reported.

For the data described in Fig. 2I we were unable to obtain a nasal wash from one of the animals due to technical issues at the time of necropsy. We have added the following line to the figure legend to identify this to the reader.

Line 374: “i, Viral RNA levels in nasal washes at 6 dpi (infectious dose, 10⁵ TCID₅₀).
Sample was obtained for 4/5 deer mice.”

Figure 3. This figure contains a lot of information and as a result is a little hard to follow. Some of the arrows are difficult to see and the claims shown are hard to justify. For example, do the green arrows point to epithelial cells in the insert of the last panel? What are the blue arrows indicating?

Thank you. We have supplied a higher resolution image of Figure 3 (the final image is very high resolution, but was too large to include in the initial submission). We also enlarged and re-adjusted the arrows to clarify. We labelled the inset panel more clearly.

Figure 4. Why is the analysis of analytes based on a higher infectious dose?

We agree with the reviewer that it would have been ideal to have performed the blood biochemistry and hematologic analyses at both doses; however, during our initial studies we did not incorporate these analyses, and fresh blood samples must be used for both assays so we could not use stored samples to perform the analysis later. In the golden Syrian hamster model of SARS-CoV-2 we have noted that a dose range of 10^4 TCID₅₀ to 10^6 TCID₅₀ resulted in a very similar disease outcome (unpublished data). We have included an additional Extended Data file (Extended Data Fig. 5) that explicitly outlines the experimental conditions used for each panel, and we believe this has improved the clarity.

Figure 5. I think it would be useful to have the experimental schematic design extended from Figure 4 to the top of Figure 5.

We agree with the reviewer that including the experimental design schematic would be beneficial; however, we were unable to include the schematic and maintain the panel sizes. We added a statement to the figure legend drawing the reader's attention to Extended Data Figure 5 that we have added with this revised version of the manuscript that outlines the experimental design.

Figure 5 Legend, Line 413: “**a-g**, Adult male and female deer mice were exposed to 10^6 TCID₅₀ SARS-CoV-2 by an i.n. route of infection (a schematic of the experimental design is provided in Extended Data Fig. 4).”

Again, what is the rationale for using the different TCID50s?

Since the initial transmission experiment in which we infected the donor deer mice at 10^6 TCID₅₀ and initiated direct contact with the naïve recipient deer mice at 24 hpi resulted in efficient transmission we sought to verify the transmissibility at a lower dose (10^5 TCID₅₀) with a delayed time to direct contact (48 hpi). The rationale was to provide better evidence of transmission by using the lower dose and delaying the time to contact to be more sure that transmission was not occurring through the spread of inoculum given to the donor mice.

Also, please define d.c. in the figure legend.

Thank you. We have corrected the figure legend.

Figure 5 Legend, Line 414: “At 1 dpi individual inoculated donor deer mice were transferred to a new cage and co-housed with a single naïve deer mouse (1:1 ratio) to assess SARS-CoV-2 transmission by direct contact (d.c.)”

Reviewer #3 (Remarks to the Author):

This is the first report of SARS-CoV-2 infection and transmission in North American deer mouse. Because the infection is asymptomatic or causes very mild symptoms, it cannot be used as the model for symptomatic human COVID-19. The significance and value of the work are therefore compromised. In addition to reverse zoonosis mentioned in the paper, whether deer mice might serve as a model for asymptomatic infection in humans merits more exploration and discussion. For example, whether and how the immune response to SARS-CoV-2 could be studied in deer mice should be determined and discussed. Some specific concerns are listed below:

We agree with the reviewer that it would be interesting to determine the why deer mice appear to tolerate a SARS-CoV-2 burden in the respiratory tract without developing signs of severe illness (as well as infection with several other viruses). Since the Ace2 receptor is predicted to efficiently bind the SARS-CoV-2 spike we agree that the immune response to infection might yield some interesting data and explain this phenomenon. We are planning further studies to address this point.

We have added the following line to the Main text:

Line 253: “Lastly, we suggest that the deer mouse model of SARS-CoV-2 infection, as a largely outbred small animal model, may prove useful for studying viral pathogenesis, particularly the determinants of asymptomatic infection, viral persistence, and transmission, and for evaluating the protective efficacy of experimental vaccines and therapeutics.”

1) The animal model is relatively artificial, using 10^5 - 10^6 TCID₅₀ for virus inoculation.

While the administered doses are comparable to those used in other studies of SARS-CoV-2 pathogenesis in small rodents (i.e. hamsters) we agree that a natural transmission event would likely be at a lower dose (and we were careful to state this in the text). We believe that the transmission studies described in Figure 5 strongly suggest that deer mice are susceptible to infection at a lower dose since the amount of shed virus detected in the donor deer mice never exceeded $10^{2.5}$ TCID₅₀/ml, but the deer mice still became productively infected.

2) Since the age of the deer mice used ranges from 8 to 32 weeks old, there will be huge difference in terms of initial body weight and immune response.

We were also interested in the effect of age on viral pathogenesis since age differences have been reported in humans and in animal models of SARS-CoV-2 infection. We found that age did not significantly affect viral titers in the tissues as described in Extended Data Fig. 1. It is

likely worth exploring why contrary to other hosts aged deer mice do not appear to be more susceptible to infection or present with greater pathology. Based on availability of animals from our breeding colony we used animals that were as closely aged as possible, in the young to mid-aged group. In the lab, deer mice lifespans can be quite long, up to 96 months with a mean of approximately 45 months) and we did not have access to significantly aged animals that would be expected to show significant age related differences in immune responses. It is possible that in age range of 2-8 months that deer mice are relatively similar in their immune responses to infection and this will be of interest to evaluate as we work with this animal model.

3) The number, age, and gender of deer mice should be described clearly in all experiments.

Thank you. We have added Extended Data Fig. 5 to clearly describe the experimental design of the various experiments and link them to the corresponding figure panels.

4) Fig 3: The resolution of the figure is too low.

We agree with the reviewer and have provided a higher resolution version with our revised manuscript, and we have a very high resolution version that we can also provide.

5) Fig 4: Statistical analysis is not sufficiently robust.

Thank you. We revised Figure 4, panel A to include “ns” for the comparisons that did not reach statistical significance. We also added statistical analyses for panels g and h and added the following statement to the Figure legend and Results sections.

Line 185: “All five deer mice infected at 10^5 TCID₅₀ had detectable serum IgG titers against mixed spike/nucleoprotein (S/N) antigen as assessed by ELISA at 14 dpi (OD 2.7 – 2.8 at 1:100) ($P < 0.0001$, student t test) (Fig. 4g), and neutralizing antibodies by 28 dpi (plaque reduction neutralization test (PRNT₉₀) 1:40 to 1:320, $P = 0.0349$, student t test) (Fig. 4h).”

Figure legend Line 409: “* = $P < 0.05$, **** = $P < 0.0001$, ns = $P > 0.05$; unpaired student t test (a-e, g-h), Mann-Whitney test (b).”

6) Fig 5: The colors for different samples are quite indistinguishable (Fig 5e, 5f and 5h). The plots for samples are also too small to be seen (Fig 5c & 5d).

Thank you. We have increased the size of the panels and/or symbols in this figure, and we believe this has improved the readability of the data considerably and made it much easier to discern the colour corresponding to the individual deer mice.

REVIEWER COMMENTS

Reviewer #1 (Remarks to the Author):

The authors have adequately addressed my comments.

Reviewer #2 (Remarks to the Author):

No additional comments.

Reviewer #3 (Remarks to the Author):

All comments raised have been satisfactorily addressed.